# Post-Intensification Poaceae Cropping: Declining Soil, Unfilled Grain Potential, Time to Act

**DOI:** 10.3390/plants12142742

**Published:** 2023-07-24

**Authors:** Geoffrey R. Squire, Mark W. Young, Gillian Banks

**Affiliations:** James Hutton Institute, Dundee DD2 5DA, UK; mark.young@hutton.ac.uk (M.W.Y.); gill.banks@hutton.ac.uk (G.B.)

**Keywords:** Poaceae, cereal, barley, wheat, soil organic matter, soil carbon, C:N stoichiometry, crop yield, nitrogen fertiliser, soil remediation, yield gap, sustainable crop systems

## Abstract

The status and sustainability of Poaceae crops, wheat and barley, were examined in an Atlantic zone climate. Intensification had caused yield to rise 3-fold over the last 50 years but had also degraded soil and biodiversity. Soil carbon and nitrogen were compared with current growth and yield of crops. The yield gap was estimated and options considered for raising yield. Organic carbon stores in the soil (C-soil) ranged from <2% in intensified systems growing long-season wheat to >4% in low-input, short-season barley and grass. Carbon acquisition by crops (C-crop) was driven mainly by length of season and nitrogen input. The highest C-crop was 8320 kg ha^−1^ C in long-season wheat supported by >250 kg ha^−1^ mineral N fertiliser and the lowest 1420 kg ha^−1^ in short-season barley fertilised by livestock grazing. Sites were quantified in terms of the ratio C-crop to C-soil, the latter estimated as the mass of carbon in the upper 0.25 m of soil. C-crop/C-soil was <1% for barley in low-input systems, indicating the potential of the region for long-term carbon sequestration. In contrast, C-crop/C-soil was >10% in high-input wheat, indicating vulnerability of the soil to continued severe annual disturbance. The yield gap between the current average and the highest attainable yield was quantified in terms of the proportion of grain sink that was unfilled. Intensification had raised yield through a 3- to 4-fold increase in grain number per unit field area, but the potential grain sink was still much higher than the current average yield. Filling the yield gap may be possible but could only be achieved with a major rise in applied nitrogen. Sustainability in Poaceae cropping now faces conflicting demands: (a) conserving and regenerating soil carbon stores in high-input systems, (b) reducing GHG emissions and other pollution from N fertiliser, (c) maintaining the yield or closing the yield gap, and (d) readjusting production among food, feed, and alcohol markets. Current cropping systems are unlikely to satisfy these demands. Transitions are needed to alternative systems based on agroecological management and biological nitrogen fixation.

## 1. Introduction

Degradation of soil resulting in loss of soil organic carbon and reduced water holding capacity has occurred worldwide due to the conversion of natural ecosystems to agriculture and forestry [1,2,3,4]. Repeated and highly fracturing tillage and trampling by livestock has diminished gains of soil organic matter through plant residues and exudates while increasing loss through disaggregation, breakdown, gaseous release, and removal by wind and water [5,6,7,8,9]. In many areas, severe degradation of soil has limited crop and grass biomass production, and hence the capacity of agriculture to regenerate soil organic matter [2].

A sustainable future for agriculture is nevertheless vital. The most recent phase of intensification, between 1960 and 1990, raised yield to support a doubling of the global population. It was achieved with little increase in arable land [10], easing the pressure to convert more forest and grassland to agriculture [11]. Poaceae crops had a central role in these developments [12,13,14]. However, several decades after the initial increase, yield has levelled or even declined in some major crops [15,16,17,18,19]. Negative feedbacks of intensification may therefore be degrading the biophysical processes in soil that support crop growth and yield [11]. More widely, intensification has led to negative effects on the environment and health, for example through increased use of nitrogen [19,20,21], which continues to pollute water [22,23] and dominate greenhouse gas emissions from arable land [24,25]. The economic cost to the environment and health of high input agriculture might now outweigh the benefits of increased yield [21]. Given this history, global bodies such as the UN’s Food and Agriculture Organisation (FAO) have called for concerted actions that include an end to converting forest and grassland to agriculture, stabilising and regenerating damaged soil, and raising both the efficiency of resource use and the achievable yield [2]. Strategies to meet future food should also extend beyond the soil–crop system to include lessening demand by limiting food waste and over-consumption [12,26]. 

In principle therefore, the ‘solution space’ and pathways to sustainability have been defined [12]. Devising a strategy for future agriculture in any region must begin by quantifying the current system in terms of its soil and yield. As a guide, three or sometimes four ‘levels’ of yield can be identified. Figure 1 summarises the position from several studies on Poaceae crops [27]. The lowest level is a basal yield for which the crops rely only on the capacity of soil and other plants such as legumes to supply nutrients. The basal yield occurred before intensification but can be approximated in fields or experimental plots receiving no inputs. The second is the present average yield which has been raised by intensification above the basal. A third is termed a technological maximum, a value that has been achieved in the field, but usually in restricted circumstances where agronomic inputs are maximised and local limitations removed. Above them is a pedo-climatic or climatic maximum derived commonly from modelling as the highest that could be achieved in the soil and climate. The overall range of yield is determined by major climatic factors. Those to the left and centre of Figure 1 have growth restricted to 3 to 6 months of the year by water or temperature (short-season crops). Those to the right are generally not so limited and can grow for 9 to 11 months (long-season). The long-season crops are higher yielding than short-season, due to the extended period for which the leaf canopy intercepts solar radiation, but they need more nitrogen and water. In these examples, the average yield, though well above the basal, is usually much lower than the technological maximum. The difference between them—the ‘yield gap’ [28,29,30,31,32]—is most commonly determined from empirical measures of average and maximum yield, compared in some studies to potentials derived from crop modelling. The gap is variously attributed to the lack, high cost, or inefficient use of inputs and the unsuitability of modern crop varieties to adverse conditions; but less understood are the negative feedbacks referred to earlier of high-input cropping on soil processes. 

The status of soil and main Poaceae crops are now examined for farming systems in a high-yielding region of eastern Scotland, UK. The approach first relates carbon acquisition and yield of crops explicitly to soil condition, defined primarily by organic carbon content. It then defines the yield gap through measuring the components of yield (number and mass of heads and grains) rather than relying on bulk measures on commercial fields. The region has a 5000-year history of agricultural land usage [33]. Land brought under tillage, which comprises much of the lowland, maritime east of the study region, experienced several phases of intensification, the most recent from 1960. Most crops before the 1960s were short-season (SS), mainly oats and barley. Intensification included deeper tillage and regular application of mineral fertiliser and chemical pesticide. From the 1980s, a proportion of the SS crops were replaced by long-season (LS) varieties that are higher yielding and given, on average, twice the inputs of fertiliser and pesticide compared to SS crops. Mean cereal yield trebled between 1960 and 2000, but then stabilised. By 2010, the main Poaceae crops, covering >80% of the tilled land, comprised both SS and LS varieties of barley (*Hordeum vulgare* L.), wheat (*Triticum aestivum* L.), and oat (*Avena sativa* L.). By global standards, soils in general are highly suitable for agriculture [34,35] but negative aspects of intensification include declines in soil quality and in-field biodiversity [36,37], risk of soil erosion [38], continued pollution of water [39], and high greenhouse gas emissions [40]. The region is well suited to the comparisons needed because not all of it was intensified to the same degree. By 2010, 50 years after intensification began, a diverse set of crops and grass existed, subject to a range of management intensity and crop carbon acquisition within the same latitudinal range and climate. 

The work reported here concentrates on the two main Poaceae crops of the region, SS barley and LS wheat [14]. Specific hypothesis examined are (1) that soil carbon ‘store’ and crop carbon flux are inversely related, (2) that main drivers and limitations to yield include soil status and applied nitrogen, and (3) that the technological maximum yield and yield gap can be quantified through measures of grain number and mass. Conclusions are presented on options to improve soil quality and yield components to achieve long-term sustainable output. The methodology and findings of the work should be generally relevant to assessing the post-intensification status of Poaceae crops in agricultural land.

## 2. Results

### 2.1. Soil Carbon and Nitrogen

The results of soil C and N analysis are given in relation to the estimated intensity of management defined by previous crops (Figure 2). The sites sampled in the study achieved the intended two-fold range of C-soil from 2 to 4%. With one or two exceptions, C was above 3% for low-intensity sites and below 3% for high-intensity sites (Materials and Methods gives definitions). The total carbon store in the upper 0.25 m of soil, derived from the values in Figure 2a, ranged from 68 to 155 t ha^−1^, expressed in Figure 2b in 1000 kg ha^−1^ to allow comparison with data on crops presented later. N-soil was closely correlated with C-soil across the whole range to give C:N ratios between 10 and 14 around an average of 12.1, shown by the dashed line.

### 2.2. Crop Carbon and Nitrogen

The procedure for selecting sites for the study resulted in short-season (SS) barley and long-season (LS) wheat being distributed across a range of C-soil. As is invariably the case in this region, crop dry matter production was greater for LS wheat than SS barley (site means: 14.02, 7.47 t ha^−1^, *p* < 0.001), while C as a % of dry matter was similar for both at around 44%, typical for living plant material. As described in Materials and Methods, differences between species in crop dry matter and C-crop were largely determined by the length of the growing season and the nitrogen available to support growth and grain protein content. 

In barley, C-crop ranged from 1420 to 4960 kg ha^−1^, and N-crop from 32 to 97 kg ha^−1^, the lower values fertilised from previous livestock grazing and the higher from targeted application of mineral fertiliser. The relation between C-crop and N-crop was highly conserved throughout the range (r = 0.95), giving a mean C:N of 52.2 (±1.64). In wheat, C-crop ranged from 4740 to 8320 kg ha^−1^ and N-crop from 80 to 282 kg ha^−1^. C-crop and N-crop were correlated (r = 0.90), but C:N ratios were more variable around a mean of 39.7 (±2.29), ranging from values similar to those in barley down to a low of 25. C:N ratios varied in relation to intended uses or markets: barley going mainly to malting and feed that require a low protein content, and wheat to more diverse markets, those with lower C:N requiring the highest protein content for quality feed or milling. Yields recorded in a government survey before intensification and in the decade before sampling were converted to equivalent C-crop values (Material and Methods) and shown by the arrows in Figure 3. The lowest few barley sites were similar to pre-intensification values, while the current means fell within the ranges of both crops indicating that many of the sites sampled were typical of current production.

No limiting effects were detected of soil carbon on crop production. For the species treated separately, relations between C-soil and both C-crop and crop C:N ratio were not significant. For the species combined, C-crop (y, kg ha^−1^) showed a weak negative relation with C-soil (x, %) (y = 8182 − 1161x; F = 7.97, *p* = 0.010), whereas that for the C:N ratio was not significant. The negative relation arose because the current high N inputs and N-crop at highly intensified sites gave high production irrespective of low C-soil, and vice versa.

### 2.3. Crop C and N as a Percentage of Soil C and N

Data in Figure 2 on soil and Figure 3 on crops are now combined in two indicators—C-crop/C-soil and N-crop/N-soil, expressed as percentages (Figure 4). The general shape of the overall response was similar to that in Figure 3. For barley, C-crop/C-soil ranged from 0.91% to 5.66%, and N-crop/N soil 0.31 to 1.21%. In wheat, C-crop/C-soil ranged from 4.90 to 11.98, and N-crop/N-soil from 1.15 to 4.88. Low values of these indicators were generated at sites where long-term, low-intensity management resulting in high soil carbon was combined with low carbon acquisition by the current crop. Conversely, high values were generated by low soil carbon at high-intensity sites and high crop carbon acquisition. The combinations of soil store and crop flux extended the range of values among sites beyond that in Figure 3. The highest C-crop was 5.9 times the lowest and N-crop 8.8 times the lowest, whereas for crop and soil together, the highest C-crop/C-soil was 13.1 times the lowest and the highest N-crop/N-soil, 15.7 times. 

The difference between sites in Figure 4 is due more to variation in the crop than the soil. Absolute values on the vertical axis are related to differences in available nitrogen that results in a wide range of annual carbon flux (C-crop) compared to the carbon store (C-soil). The values of N-crop/N-soil are smaller than those of C because of the ability of relatively small quantities of applied N to generate high rates of carbon assimilation for canopy growth, physical support, and dry matter bulking. The ratio of the axes in Figure 4, shown by the dashed lines, is also indicative of the generally weak capacity of the crop to generate matter with a C:N ratio comparable to that of soil. The ratio of C-crop/C-soil to N-crop/N-soil was 4.06 (±0.19) for barley and 3.22 (±0.205) for wheat. Most sites were positioned between 5, resulting from a combination of a crop C:N 60 and soil C:N 12, and 3, produced by crop C:N 36 and soil C:N 12. If crop dry matter were used to boost soil carbon, then these ratios indicate the degree to which crop matter would need to be adjusted (by microbial action) to remove C in preference to N. The implications of these ratios for soil status and regeneration are discussed later.

### 2.4. Variation and Limitation in Yield

The analysis now moves on to consider the second group of sustainability pathways—the factors limiting yield and the yield gap. To this point, crop matter has been expressed through C and N to allow direct comparison with soil organic matter. Yield is generally expressed in units of plant dry matter of which carbon constituted around 44%. For comparison, a value of 4000 kg ha^−1^ for C-crop in Figure 3 is raised to 9090 kg ha^−1^ dry matter. Two other attributes are used when converting between C-crop and yield: the fraction of dry matter in heads; the harvest index, which was highly conserved among sites and species at 0.62 (±0.012) in barley and 0.61 (± 0.01) in wheat (ns); and the corresponding fractions for nitrogen, 0.84 (±0.006) in barley and 0.83 (±0.010) in wheat (ns). Typical of modern cereals, therefore, N was more concentrated in the head than in the whole plant, resulting in systematic reduction in head C:N ratios to 36.6 (±1.00) in barley and 28.2 (±1.37) in wheat. Conversely, the C:N of stem and leaf was higher than that for the whole plant, the mean for barley 132 (±6.5) and wheat 97 (±12.2).

Head dry matter per unit field area ranged widely among sites from 2.35 t ha^−1^ in the lowest yielding barley to 13.23 t ha^−1^ in the highest wheat, a 5.7-fold difference similar to the 5.9-fold range in C-crop in Figure 3 and consistent with the conserved harvest indices cited above. Means for species were 4.56 t ha^−1^ (456 g m^−2^) in barley and 8.63 t ha^−1^ (863 g m^−2^) in wheat (*p* < 0.001). This variation in head dry matter is now examined in terms of the components of yield. To avoid the large values when expressed per hectare, stem and grain number are expressed per square metre (with conversion given as necessary). First, the two components, stem number per unit area and individual head mass, are shown in Figure 5 in relation to ‘yield contours’, each of which defines the variation in the two attributes for a given head dry matter from 2 to 12 t ha^−1^ (200 to 1200 g m^−2^, 2000 to 12,000 kg ha^−1^). The species differed mainly in individual head mass: 0.94 g in barley and 2.1 times larger at 1.97 g in wheat (*p* < 0.001). Among sites, stem number in barley varied more widely (300 to 740 m^−2^) than individual head mass (0.78 to 1.20 g) and strongly determined head dry matter. Regression indicated the latter as increasing by 96 g m^−2^ for every 100 heads (y = 0.96x – 0.39, F = 34.3; *p* < 0.001). However, individual head mass was not without effect—the highest total head mass (above the 6 t ha^−1^ contour) occurred with intermediate stem number and maximum individual head mass. In wheat, the relation between number and total head mass was weaker (y = 1.34x + 273; F = 5.77, *p* = 0.035) but large variation also occurred in individual head mass. 

Dissection of heads showed that, of the two components of individual head mass, number of grains was a stronger determinant than individual grain mass. The species differed significantly (*p* < 0.001) in grain number per head, 22.7 (±0.38) in barley and 39.6 (±1.22) in wheat, 1.74-fold higher in wheat. In contrast, individual grain mass was similar (ns) at 41.5 mg (±0.82) in barley (hulled grain, awns removed) and 41.3 mg (±1.63) in wheat (naked grain). Total head mass was therefore strongly related to the number of grains per unit field area, calculated as stem number multiplied by grains per head. Grain number ranged among sites from 6190 m^−2^ in low input barley to 24,140 m^−2^ in high-input wheat. The relation between grain number and head dry matter differed only slightly between barley and wheat (Figure 6). Arrows to the right in Figure 6 show N-crop at sites of minimum and maximum head mass, and for groups of sites close to the regional averages. Data on grain number are not available from the official crop census, but the implication from Figure 6 is that yield has been raised in the 50 years since intensification began by major increase in grain number per unit field area driven by nitrogen and related management.

### 2.5. Estimation of Yield Potential from the Reproductive Sink

The inference from Figure 6 is that—within each species—yield over the range of conditions studied is not genetically sink-limited by the number of grain sites that can be produced per unit area. Instead, yield is limited by the amount of carbon (resource) that is assimilated and partitioned to the reproductive sink. In turn, the carbon assimilated is determined by the intensity of management, primarily nitrogen availability (Figure 3, Figure 4 and Figure 6). Where more resource is available, the crop produces more reproductive heads and fills more grains per head. (Unfilled grain sites were defined as those where a spikelet was discernible but had no grain, or else in wheat where florets on spikelets were discernible but had no grain). Deviation of sites from the fitted lines in Figure 6 indicates where individual sites also differed in mean grain mass. 

Measured variation in components of yield within sites and site sampling loci is now exploited to demonstrate the degree to which typical yield could be raised if a crop contained more or heavier grains. First, a set of sites were selected for each species for which total grain dry matter was close to the regional means from government survey and which excluded sites with more extreme characteristics (e.g., high or low head number per unit area). The resulting sites for barley had head numbers of 580 m^−2^ and total head mass of 6.13 t ha^−1^ (compared to the official regional mean 5.75 t ha^−1^) and for wheat 470 m^−2^ and of 8.76 t ha^−1^ (regional mean of 8.29 t ha^−1^). Heads dissected from these sites (216 in barley, 144 wheat) were ranked in order of individual head mass and several upper percentile ranges selected (30, 20, 10, 5%), for which grain number and individual grain mass were recalculated. The yield measured on all sampled heads and raised yields are linked by the two vertical lines in Figure 7. 

The measured yield is located at the low point of the line and the corresponding grain number and individual grain weight shown in the adjacent box together with the actual N-crop measured (as in Figure 3). The upper four bars on the line show the corresponding grain number and individual grain mass of the top 30, 20, 10, and 5% of heads. The two species had a similar individual grain mass and differed mainly in grain number. Boxes for the top 30% and 5% show both grain number and mass increased as the selection narrowed. The boxes also show the whole-plant N-crop that would occur at the raised yield. Two values are given for raised yield at different C:N ratios: 30 and 35 for barley and 27 and 23 for wheat (Figure 3). For example, if all heads were raised to the characteristics of the top 30%, wheat would have a yield of 13.1 t ha^−1^ (raised from 8.76), heads would have a mean of 51 grains of 48 mg individual mass, and the total N-crop required to generate this would be 258 kg ha^−1^ at C:N 27 and 303 kg ha^−1^ at the higher protein content of C:N 23. The contribution to raising yield at all four percentile ranges (Figure 8) is similar for number and individual grain mass in barley but occurred mainly through increased grain number in wheat (mainly through filling more florets per spikelet). That raising yield above the average in this manner is in principle realistic is shown by the highest-yielding wheat site (excluded from this analysis), for which yield was similar to the 30% raised value, for which grain number per head was 49, individual grain mass 46 mg, and measured N-crop at 282 kg ha^−1^ and head C:N ratio 23.9.

## 3. Discussion

The contributions of the main Poaceae crops in this region to food and other products have changed over past centuries and even over recent decades. Production among the species, mainly barley, wheat, and oats, has varied between food, livestock feed, and alcohol [41]. Oats once dominated the sown area, but now barley and wheat are more prevalent. Most food was once produced locally but after 20th century intensification, cereals were grown mainly for alcohol and feed. Short-season crops dominated before intensification, but now long-season wheat and barley contribute 40–45% of total grain production. Change is certainly possible but whatever the future of cereal cropping, success will depend on resolving the conflicts between regenerating soil, reducing the nitrogen footprint of agriculture, and managing the yield gap. 

### 3.1. Regenerating Soil and Reducing the Nitrogen Footprint

The main obstacle to the regeneration of soil organic matter in many parts of the world is a lack of nutrients and organic inputs [2]. Such soils have degraded to the point where their condition is severely limiting carbon acquisition by plants [8]. In the region studied here, sites of low C-soil and high management intensity are not yet severely degraded [35]. They would be recognised in FAO accounts as needing action before more serious decline occurs. The ranges of soil carbon and crop production of the sampled fields were determined more by choice to pursue a particular type of farming and management intensity than by physical factors such as climate or topography. The type and intensity of farming could therefore be modified. Moreover, soils at the lower end of the soil carbon range have not lost a capacity to support high crop carbon acquisition of 15–20 t ha^−1^ dry matter annually.

However, regeneration of soil is unlikely through carbon acquisition from current high-input crops. Returns of C to soil have become dependent mainly on annual production in roots, since most stem and leaf matter is removed at harvest for livestock feed. Root production was not measured here but is typically an additional 20–30% of production above-ground [42,43]. Yet continued deposition of root matter in the several decades since the rise in LS crops has not prevented low soil carbon. Even if stem and leaf were re-incorporated, these vegetative residues now have very high C:N ratios due to the historical rise in the harvest indices for dry matter and nitrogen [14,44]. For example, soil at 2% C contains 65,000 kg ha^−1^ C in the upper 25 cm (Figure 2). Regenerating this soil layer to 3% C is equivalent to the addition of 32,500 kg ha^−1^ C and 2700 kg ha^−1^ N at a C:N ratio of 12. In contrast, stem and leaf of the average LS wheat held 2380 kg ha^−1^ C and 28 kg ha^−1^ N. A +1% change C-soil is therefore equivalent to 13–14 times the annual acquisition of C in stem and leaf and around 90 times the annual acquisition of N. Further interventions are needed therefore to regenerate soil carbon at high-intensity sites. The principles behind building more stable soil carbon fractions are well understood [45,46] and remediation is achievable by a range of practices including the substitution of grass or legumes in a cereal rotation, reduced tillage, and external carbon amendments [47,48,49,50,51]. Some crops such as oats, a legume, or a mix of the two have been also shown to increase the yield of a subsequent cereal [52]. 

Similarly, major change can be brought about to reduce wastage of applied N and the GHG emission equivalent of mineral fertiliser. In terms of efficiency, defined as uptake/applied N, the high yielding wheat crops sampled had a mean efficiency of 0.91, while the fields showing very high production with >200 kg ha^−1^ N fertiliser had efficiencies around 1. Caution is needed in applying this comparison more widely since the year of study supported high production and yield due to favourable weather. More generally, uptake/applied efficiency in major cereals is low in intensified agriculture [53] and has been much reduced in the study region by prolonged wet weather, leading to uptake/applied N in wheat below 0.5 [40]. Moreover, such high rates of applied N compromise any attempts to reduce overall GHG emissions from arable land [40]. 

Given the complexity of nitrogen reactions and pathways in soil [54], losses of mineral N are best reduced through what is termed a ‘portfolio approach’ to low nitrification management, in which a range of interventions are introduced to cover for unexpected variation in the growing environment [20,55,56]. Interventions include diversifying cropping systems, minimum tillage, applying nitrification inhibitors, and perhaps most effectively, introducing a grain or forage legume in a cereal rotation [57]. At an experimental field platform supporting a high input rotation in this region, uptake/applied N measured over a run of several years was raised above 1 by a combination of such practices, but particularly by the capacity of a grain legume, field bean, to accumulate around 200 kg ha^−1^ N mainly by biological fixation [40,58]. In principle, regenerating soil carbon, reducing nitrogen loss, and meeting GHG emissions targets are feasible in this region and could be achieved given major change in choice of crops and agronomy, but particularly by raising the area grown with legumes.

### 3.2. Managing the Yield Gap 

Closing the yield gap is more problematic and requires further examination beyond raising the efficiency of applied resource use. Source-sink relations in cereals can be complex [43,59]. The source (usually taken to be photoassimilate) and the sink (e.g., number of grain sites) interact such that a limitation in one can cause a limitation in the other. Characteristics of individual heads may be influenced by competition between heads for resources; and within heads, there may be trade-offs between grain number and mean grain mass. The results here indicate an overall limitation of carbon acquisition and yield operating mainly through applied nitrogen. Crops at the low end of the production range yielded similarly to cereals in the two decades before intensification and represent a basal yield. In barley, more applied nitrogen produced more heads per unit area, but the decline in mean head mass at the higher limit of head number suggests insufficient resource was shared between the many heads. In wheat, head number was less of a discriminating factor because of the much greater capacity to set more grains per head. Again, more nitrogen applied led to more grains set and filled overall. The great variation in head mass, grain number per head, and to a lesser extent mean grain mass (Figure 7 and Figure 8) is therefore likely due to resource limitation: heads and potential grain sites may be determined early in phenology but then do not fill to capacity. 

The detailed analysis of grain number and mass showed that yield could in principle be raised substantially in both species if all heads were filled to the same degree as the top 30%, 20%, etc. (Figure 7 and Figure 8). However, the yield of two-rowed barley could not be raised to the upper range of yield shown by wheat. Two-rowed barley was source-limited among sites in this study but sink-limited when compared to the higher measured and projected yield of wheat. The overall increase in cereal output during intensification could not have been achieved by improvements to spring crops alone. It needed the increase in winter crops with their higher grain sink. A full analysis of grain phenology is not feasible here, but the variation in head mass in wheat was due mainly to the number of florets (each with a grain) per spikelet. The capacity for increase in two-rowed SS barley was much less than in LS wheat (Figure 8) because of the limited number of grains per head (1 grain per spikelet in barley and up to 5 in wheat). Further work is needed to assess whether the higher potential grain number in 6-row barley [60] would result in a higher grain sink. In both crops, however, narrowing the yield gap would lead to a very large rise in applied mineral nitrogen (Figure 7). Success would also require that the application of nitrogen sufficiently promoted high leaf area, canopy duration, solar interception, and carbon acquisition to generate the assimilate needed. In the context of long-term cereal breeding [14,44,61], emphasis should now be on traits for more efficient resource capture.

Decisions of whether and how to reduce the yield gap should be part of wider discussion. The overall production gap in this region, defined as current agricultural output in relation to needs, is uncertain and has to be resolved. Since only around 7% of the cereal crop, mainly oats, is used directly for milling for human consumption, the region relies heavily for food on imports of Poaceae products, particularly wheat [62], which makes it vulnerable to external shortages due to climate, trade, or geopolitical impacts. Some of the proposed strategies for narrowing the production gap in any region or country [12] are not feasible here. For example, the intensification of land by relay cropping as practiced in some countries is not possible for cereals because the cold winters allow germination either in autumn or spring but not both in the same field. Moving land between grass and arable is a more realistic option: of the total arable and managed grass, arable is now 30% but was previously as high as 50%. Whichever pathways are taken, the varied cultural attitudes to farming need to be respected. Sites of low crop assimilation and yield (Figure 3, Figure 4, Figure 5 and Figure 6) are not low because of limitation by climate or soil, but because farming there wishes to build and maintain soil carbon and use little or no mineral nitrogen. Important lessons for the future can be learned from practices across the whole range of soil condition and plant production. 

Flexibility in managing the yield gap may also be possible through manipulation of botanical traits. The dominant aim in mainstream agriculture is to maximise grain number per unit field area, but strategies to achieve this fall short: at average current yields, most heads in both species were not ‘full’. Therefore, maximising yield per plant may offer greater flexibility: yield could be maintained, or even increased, by increasing mean head mass at the expense of head number. In mixed cropping or intercropping for example, the additional field ‘space’ would be occupied by other species such as grain legumes that would also reduce the addition of N fertiliser. Ultimately, a transition in cropped area and choice of products grown will need societal and political agreement on how to redirect agricultural support [63].

## 4. Materials and Methods

### 4.1. Study Region and Crops

The region of arable-grass cropland to the east of Scotland is the subject of long-term ecological study [33,36,37]. Following 50 years of intensification (see Introduction), the main Poaceae crops are now short-season barley (*Hordeum vulgare* L.) and long-season wheat (*Triticum aestivum* L.) grown for specific supply chains, including distilling, brewing, grain for livestock feed, whole-crop livestock feed, and to a lesser extent grain for milling. Not all fields were intensified to the same degree such that by 2000–2010, the tilled land supported a range of cropping systems (Appendix A): fields that had been heavily intensified, supporting mostly (LS) high-input winter crops; fields that have been less heavily intensified, growing mainly SS crops, some with grass leys; and fields growing a mix of LS and SS crops in sequence [64]. The SS and mixed systems also include a proportion of farms that by choice use no mineral fertiliser or pesticide. The major crops are not unique to either low or high input sectors—for instance, SS spring barley dominates the intermediate and low ranges but also occurs occasionally in sequence with LS crops in high-input fields. The time of field sampling for the present study, 2013–2014, was therefore 50 years after the beginning of intensification and 30 years after the rise in area of high-intensity LS crops (see Introduction). Yields in lowland Scotland from an official government survey [65] are presented for comparison with data from the sampled fields. Pre-intensification yield is that recorded for all cereals (mainly SS) in the period 1940 to 1959. Current yields of barley and wheat are averaged for the ten years up to 2014. Depending on which attributes are compared, yields are converted to total above ground dry matter using a measured or estimated harvest index and then to carbon content using %C values for crops (as described later). 

Agronomic inputs to SS barley are lower than to LS wheat (Appendix A). Of the three main fertilisers, phosphate and potash are applied in similar quantities to both crops but are omitted in some years. In contrast, nitrogen is applied annually to >98% of fields to achieve the target yields of grain or forage (Appendix A). In the government fertiliser census for 2014, mineral N fertiliser averaged 127 kg ha^−1^ across the arable sector as a whole, 179 kg ha^−1^ in LS wheat and 106 kg ha^−1^ in SS barley [66]. However, the range of N input varies widely from field to field, depending on economic returns and intended usage, values in government statistics ranging from zero to 225–250 kg ha^−1^ among surveyed fields for the main cereals in 2014. 

### 4.2. Crop Sequence and Nitrogen Inputs

Fields were sampled within the east Scotland region defined previously [66] between latitudes 55.5 and 57 N, lying within a zone of moderate, oceanic climate receiving typically 3.3 GJ m^−2^ solar income per year. The aim of sampling was to obtain a wide combination of soil and crop characteristics. Fields were identified covering ranges of (1) general cropping intensity, from high-input arable to low-input arable-grass, based on cropping history recorded in previous studies, and (2) likely high and low carbon acquisition during the current year, achieved by high-input LS wheat or moderate-input SS barley. The aim was to sample sites that ranged from a high soil carbon store and low crop carbon flux to those of low store and high flux. A condition was that fields were within commercial management; none were on research stations or experimental platforms. Initially, 50 fields were identified, for which farmers were asked to provide information on previous crops and likely crop management in the current year. Fields were then categorised based on typically 12 previous cropping years into low and high intensity groups—the low consisting of SS arable in sequence with grass, SS arable only, and SS and LS mixed; the high-intensity mostly LS sometimes with crops such as potato in previous years that receive very high inputs and intrusive tillage (further information in Appendix A). Sites were visited while crops were in the vegetative phase (Appendix A) to validate crop type, general management intensity, and field characteristics. Twenty-five fields were then chosen for detailed sampling to cover the ranges of cropping history (similar numbers of low and high intensity) and current crop, comprising similar numbers of LS wheat (most grown for livestock feed or alcohol, some for milling), and SS barley (for livestock feed and alcohol). The crop grown in any field was entirely the choice of the farmer, not imposed or influenced by the research project. Yield and N content were not known at the time of sampling.

Farmers were asked to provide details of the fertiliser applied to fields in the year of sampling. Mineral N ranged from zero in some low-input arable-grass to 270 kg ha^−1^ in wheat, comparable to the range found in the government fertiliser survey [66]. Nitrogen inputs proved difficult to estimate from data provided for some fields, especially those of low or intermediate input that used a combination of livestock manure and mineral fertiliser or only livestock manure. Nitrogen in the manure could not always be estimated reliably from information provided. For example, some fields of low-input arable-grass did not apply manure directly to the current crop but relied on manure dropped by livestock on the fields in the previous year. Therefore, the main discriminating variable for all sites and crops was taken to be the nitrogen content in the plants. Data from farmers were compared with government surveys (cited earlier) to check that fields were representative of the different farming systems in the region. 

### 4.3. Sampling and Processing Soil and Plants

Samples of soil and plants were taken at three points (loci) along three transects running over 100 m into a field, in total nine sample loci per field. Soil at each locus was mixed to a depth of 0.25 m, and a sample taken, dried in the laboratory, and processed for %C and %N by weight through an Element Analyser, as in previous studies [40,67] and adjusted by soil bulk density [36] to estimate C and N mass per unit field area, termed here C-soil and N-soil, respectively. These measures are not intended to estimate total C and N content of the soil, but C and N the layer of soil that is routinely tilled and fertilised, and in which there is active germination and root growth. 

Sites were visited several days before the farmer’s harvest to sample mature crop and associated weed material. Quadrats of 0.25 m^2^ were positioned at each of the 9 sample loci, the number of crop stems within the area was counted and plant material bagged for processing in the laboratory to obtain dry matter of reproductive (heads) and above-ground vegetative (stem and leaf) structures. Samples were processed for %C and %N (as for soil described earlier) from which C and N mass per unit field area (C-crop and N-crop) was derived from dry matter. Weeds were sampled where present but constituted very little of the total matter. In many of the LS crops, for example, no weed matter was found in any of the nine quadrats. Weeds were therefore not included in the subsequent analysis of plant production. 

To obtain the components of yield, cereal ears (six or more per locus) were removed from each sample before drying and dissected to record the potential number of grain sites per ear, determined by the number of spikelets and florets, the actual number of grains per ear, and mean grain weight. (Grains of the two species are shown in Appendix A.) Data from sample loci were combined to give per site values of the number of grains per unit field area, calculated as number of head-bearing stems per unit area and the number of grains per head. To estimate the degree of sink-limitation and the potential for increase in yield, a sub-set of sites were chosen as being typical of the crop and similar in yield to the national average provided by government survey. Heads from these sites were first ranked by mass, and then averages of head mass, grain number, and mean grain mass were obtained for all heads and stated upper percentile ranges (30%, 20%, 10%, 5%). Mean yield was then raised by the ratio of the yield in the stated percentile range to the mean yield. The N-crop likely to be associated with the raised yields was then estimated based on assumed harvested fractions of dry matter and N and C:N ratio for grain typical of the crop species and a specified grain market. 

The primary aim of analysis was to quantify relations among soil and various crop attributes, rather than compared SS and LS crops. Primary (measured) attributes such as C, N, dry matter, yield, plant density, grain number, and mean grain mass were determined at each sample locus. Values for the 9 loci at each site were averaged and presented with standard errors (±SE) to indicate within-site variation. Secondary (calculated) attributes such as C:N ratio and C-crop/C-soil ratio were similarly determined from two or more measured attributes at each sample locus and averaged per site (±SE). Relations among sites were examined in several ways [68]. A correlation coefficient (r) was estimated to show the degree of association between attributes. Linear regression was used to quantify the dependence of one attribute on another. For crops, the main determining variable was taken to be nitrogen, since its variation among sites resulted from agronomic inputs that were intended to achieve the target value of dry matter, yield, carbon content, plant population, and grain number. Linear regressions are presented with slope, intercept, F-statistic (higher value indicating more of the variation is explained by the regression model), and probability of significance, *p*-value. Where appropriate, means for LS and SS crops were compared by standard difference tests for small samples, giving a *p*-value [68].

## 5. Conclusions

The procedure for selecting fields captured a very wide range of soil carbon and crop production in the region’s commercial agriculture. The measures of C, N, and C:N ratios enabled a common approach to comparing soils and the contrasting crops, short-season (SS) barley and long-season (LS) wheat. Fifty years of intensification has resulted in major trade-offs between soil and crop: sites growing mainly SS crops of low to moderate intensity, some with grass leys, had higher soil carbon stores (3 to >4%C) than those that intensified to high-input LS crops (<2 to 3%C). Across the range of soil carbon, crop carbon acquisition and yield were determined mainly by choice of SS or LS, nitrogen fertiliser, and related agronomy. Soil had not degraded to the point where it was limiting crop responses to nitrogen, but further declines should be prevented. The comparison of C:N ratios in soil and crops illustrated the difficulty in depositing more C and N to soil in current cropping systems.

Grain yield among crops and sites was determined mainly by grain number per unit area, the result of variation in head number per unit area (a stronger determinant in barley), and grain number per head (stronger in wheat). Generally, yield was resource-limited in that the grain sink was partly unfilled even at high-yielding sites. However, any increase in yield above the current average would need large increases in available nitrogen. The cropping systems are now at a crucial point. The main conclusion from the present work and related studies [33,36,40,64,67] is that conflicts between restoring soil carbon, minimizing the environmental footprint of nitrogen fertiliser, and maintaining yield or closing the yield gap are not resolvable in current, highly intensified cereal production systems. Major transitions to regenerative practices are needed, in particular the inclusion of more legumes and grass in rotation or as mixtures with cereals.

## Figures and Tables

**Figure 1 plants-12-02742-f001:**
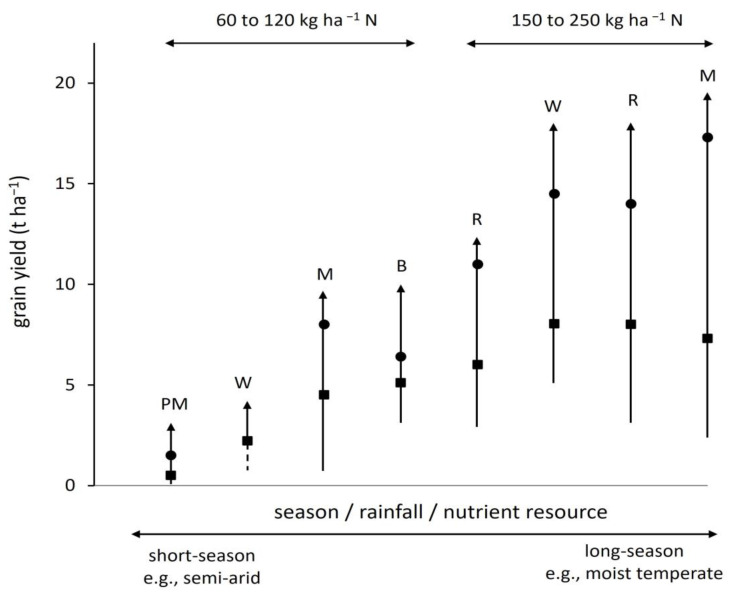
Yield ranges collated from studies of annual cereal crops of pearl millet (PM), wheat (W), barley (B), rice (R), and maize (M), shown as arrows with basal yield (lower limit), mean (square), technological maximum (circle), and pedo-climatic maximum (point); typical N fertiliser ranges to achieve the mean yield are indicated above; adapted from analysis of several independent field studies, dashed line indicates uncertainty in the basal [27].

**Figure 2 plants-12-02742-f002:**
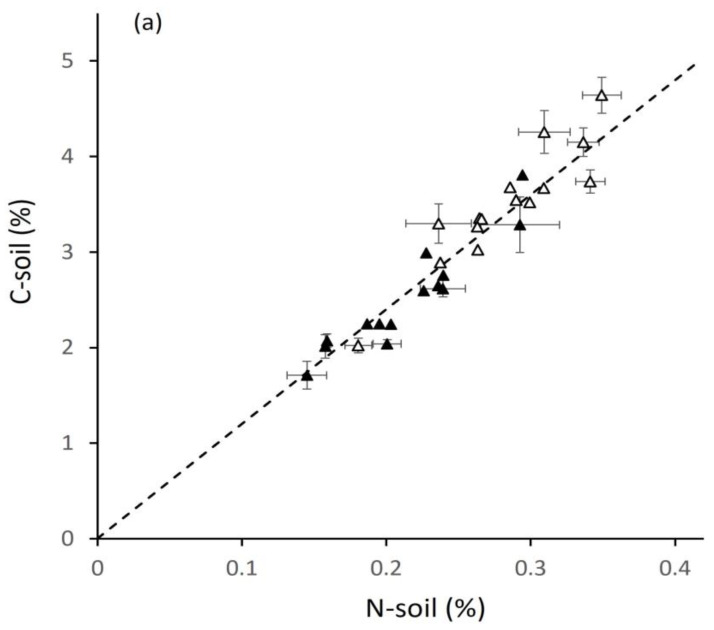
Relation between soil organic carbon (C-soil) and soil nitrogen (N-soil) expressed as (**a**) % dry soil mass, r = 0.94, and (**b**) actual mass in the upper 25 cm of soil, r = 0.89, ±SE to show within-site variation; previous cropping intensity identified as low (open symbols) or high (closed); dashed line at mean C:N of 12.1.

**Figure 3 plants-12-02742-f003:**
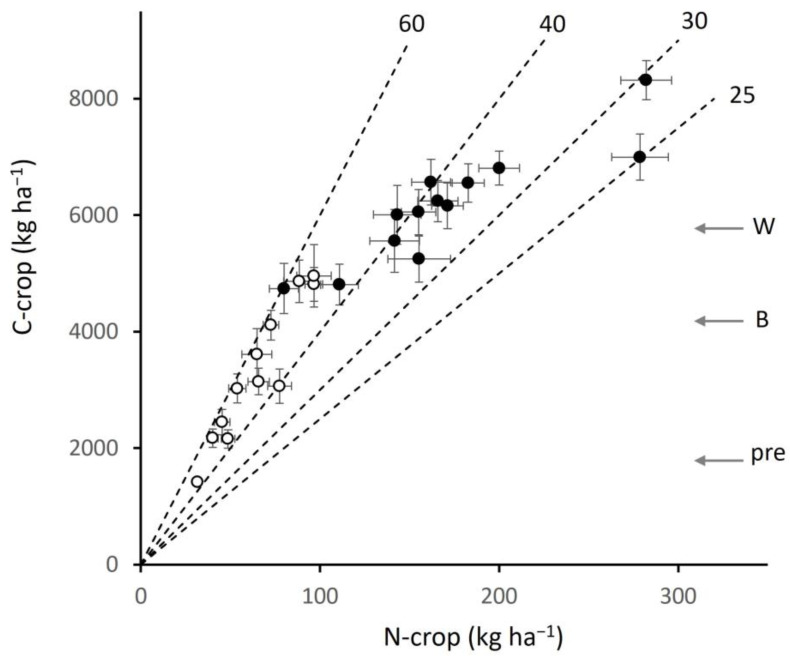
Crop carbon (C-crop) and nitrogen (N-crop) contents in barley (open circles, r = 0.95) and wheat (closed circles, r = 0.90), ±SE, dashed lines giving C:N ratios for guidance; horizontal arrows to the right showing the C-crop equivalent of regional production pre-intensification (pre), and current for barley (B) and wheat (W).

**Figure 4 plants-12-02742-f004:**
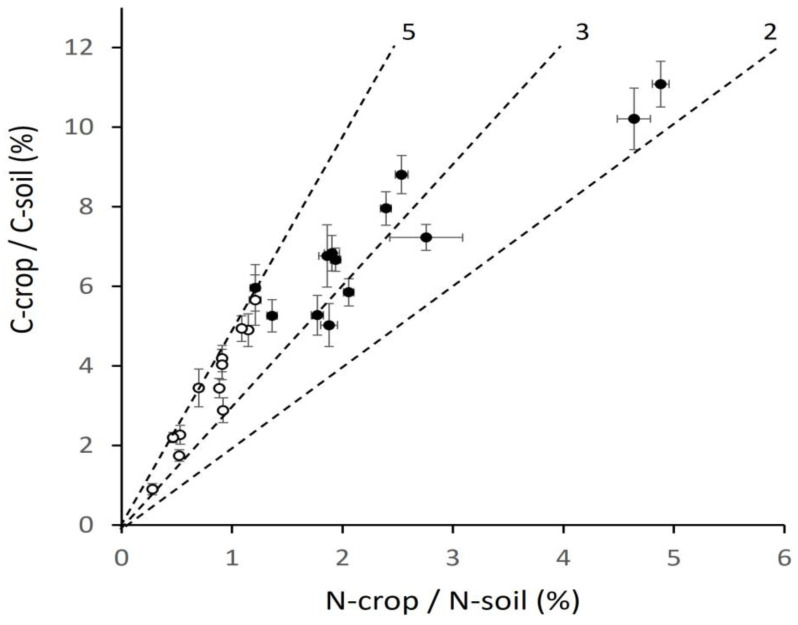
Carbon and nitrogen in crop (C-crop, N-crop) as a percentage of carbon and nitrogen in soil, upper 25 cm (C-soil, N-soil), ±SE, for barley (open circles) and wheat (closed circles), dashed lines for guidance showing ratios (5, 3 and 2) of vertical to horizontal axis.

**Figure 5 plants-12-02742-f005:**
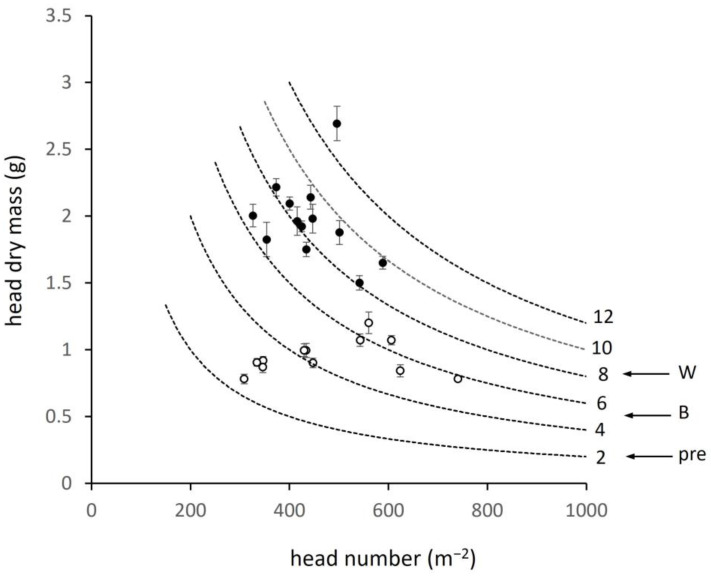
Mass of individual reproductive heads in relation to head number per unit field area for barley (open circle) and wheat (closed circle), ±SE; dashed curves indicating relations between the two variables at values of total head mass per unit field area indicated by the numbers to the right (e.g., value of 6 is 6 t ha^−1^ or 600 g m^−2^); horizontal arrows to the right indicating equivalent regional survey data for pre-intensification yield (pre) and current mean yield for barley (B) and wheat (W).

**Figure 6 plants-12-02742-f006:**
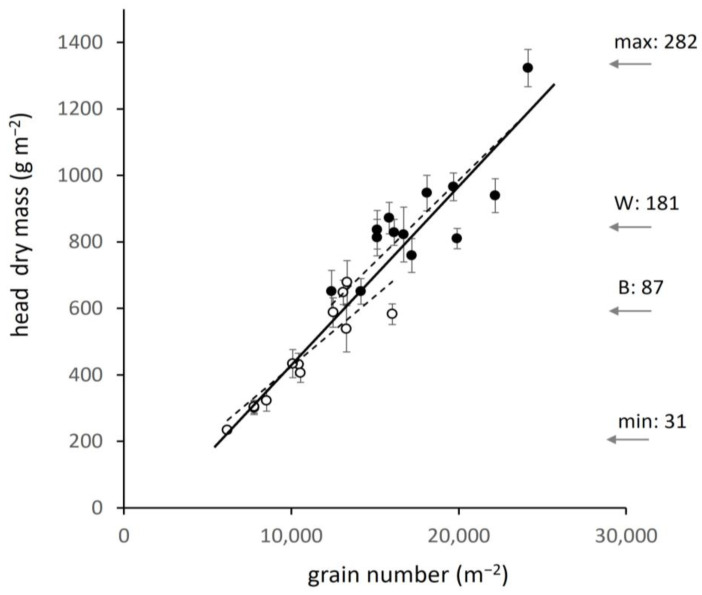
Head dry mass per unit field area in relation to grain number in barley (open circles) and wheat (closed circles), ±SE. Regressions (dashed lines): barley, y = 0.047x − 50.7, F = 53.3, *p* < 0.001; wheat, y = 0.43x + 112, F = 26.1, *p* < 0.001). Arrows indicate N-crop content (as in Figure 3) for minimum and maximum sites and representative means for barley (B) and wheat (W).

**Figure 7 plants-12-02742-f007:**
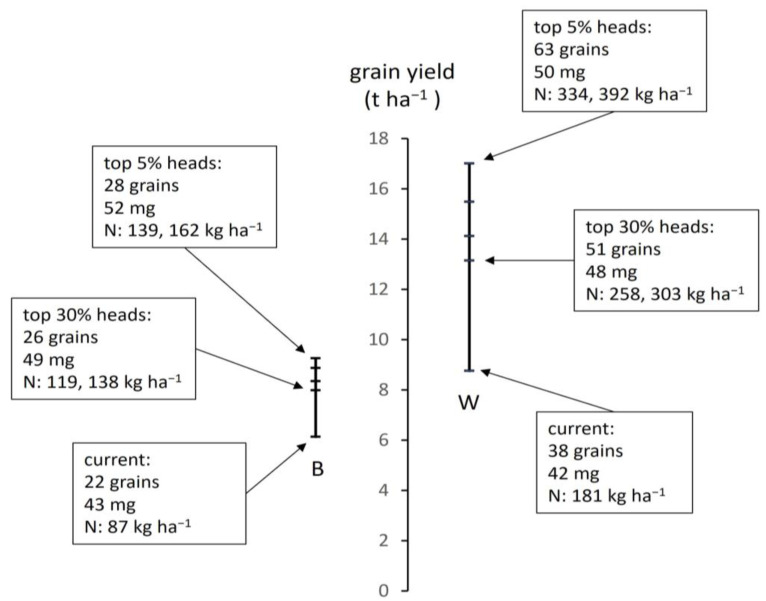
Grain yield and components for groups of barley (B) and wheat (W) sites: vertical lines linking average yield based on all sampled heads (lower bar) and the corresponding raised value recalculated for the top 30%, 20%, 10%, and 5% of heads (the four upper bars); boxes showing for current, raised to 30% and raised to 5% yields, values of mean grain number per head, mean individual grain mass and the associated nitrogen content of the head calculated for different grain C:N ratios (see text).

**Figure 8 plants-12-02742-f008:**
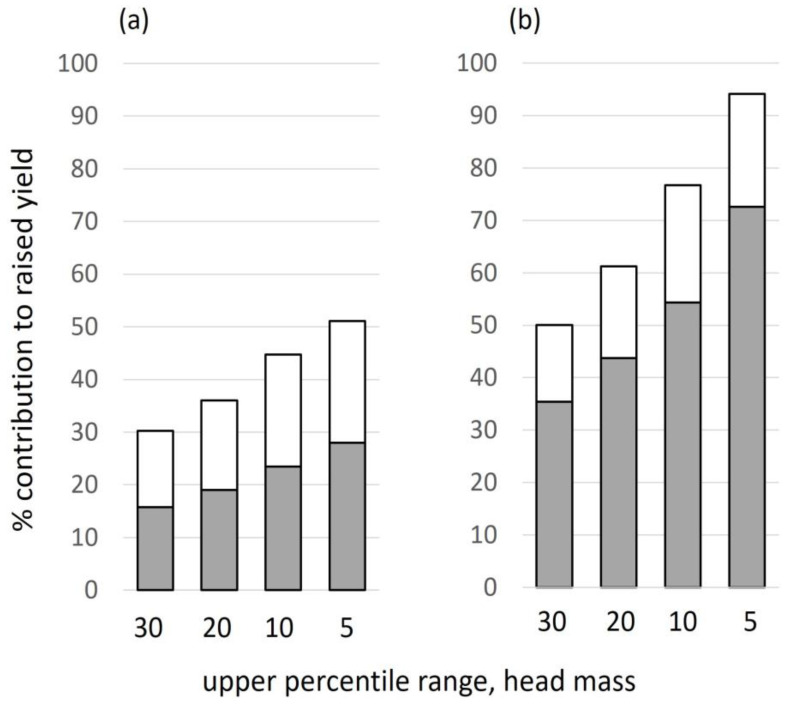
Contribution to raised yield in upper percentile ranges in Figure 7 of grain number per head (shaded part of bar) and individual grain mass (open part) for (**a**) barley and (**b**) wheat.

## Data Availability

Original government data used to generate context and regional summaries are available for download at the following web sites: crop areas and yield [65], fertiliser application [66], and pesticide [69]. Data derived from field sampling and laboratory processing are part of the extensive East of Scotland field database and available from M.W.Y. on request.

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
