# Peer review of "Post-Intensification Poaceae Cropping: Declining Soil, Unfilled Grain Potential, Time to Act"

_plants, 2023, doi:10.3390/plants12142742_

Round 1
Reviewer 1 Report
Dear authors and editor,
Thank you for the opportunity to review the paper entitled „Post-intensification Poaceae cropping declining soil unfilled grain potential, time to act„
Overall, this paper is writing up in a professional scientific way. It will be accepted after minor revision. The authors have a sound knowledge of theoretical sciences and choose a very topic that has significance.
Introduction It is well done and describes the current state of the research. In the Material and method, the section needs minor revision to add the significance of the paper.
To explain the method of collecting data from farmers and It is necessary to include the statistics used.
Best regards
Author Response
Reviewer 1
Dear authors and editor,
Thank you for the opportunity to review the paper entitled „Post-intensification Poaceae cropping declining soil unfilled grain potential, time to act„
Overall, this paper is writing up in a professional scientific way. It will be accepted after minor revision. The authors have a sound knowledge of theoretical sciences and choose a very topic that has significance.
Introduction It is well done and describes the current state of the research. In the Material and method, the section needs minor revision to add the significance of the paper.
To explain the method of collecting data from farmers and It is necessary to include the statistics used.
Best regards
The authors thank you for reviewing the paper and for suggestions on how to improve the Methods section.
Methods has been checked thoroughly and revised where needed to clarify the procedures.
Collecting data from farmers. The position in the UK is that farming is not obliged to submit statistics of aspects such as yield, fertiliser and most agronomic operations. Farmers who contribute to government surveys or allow access to their fields for scientific research do so of their own volition. The James Hutton Institute and partners has over several decades established a set of fields and farms that allow such access. For any specific piece of research, the aim is to select a set of farms that are likely to cover the driving variables in question. The fields used in the present research were selected to cover target ranges of soil condition and yield. The farms were spread across ‘types’ established in previous work cited, e.g. cereal plus grass, mainly spring cereal, mixed cereal, mainly winter cereal (now described in a new Table S1 in Supplementary Material) but the actual data were not known until after sampling. All farms were asked to supply further information on the crops grown in the fields in previous years and also on the fertiliser applied in the sampled year. This data received from farmers was checked to ensure it was representative, but could not be subject to any statistical testing. We have amended the Methods to give the rationale behind selecting sites and the comparison of our data with national statistics to assess whether the selection is representative (which is shown on some of the figures).
Reviewer 2 Report
I hope this message finds you well.
Having carefully reviewed your work, it is evident that your findings hold significant value and have the potential to make a meaningful impact in your field. Furthermore, your research not only contributes to the existing body of knowledge but also offers practical solutions and recommendations that can be implemented to address specific challenges or improve current practices.
I recently had the opportunity to read the manuscript, and I wanted to provide you with some feedback regarding its clarity and comprehensibility. While reading the abstract, I found that the pattern of writing used made it quite challenging to grasp the overall theme and purpose of your paper. The language and structure employed seem to create a barrier to understanding, hindering readers from fully comprehending the essence of your work.
With the aim of improving the accessibility of your research, I would highly recommend revisiting the abstract and considering a simpler and more straightforward writing style. By presenting the main ideas and concepts in a clear and concise manner, you can ensure that readers can easily grasp the key points and understand the significance of your findings.
Remember that effective communication plays a crucial role in disseminating knowledge, and simplifying complex ideas can greatly enhance the impact of your work. By editing and describing the abstract in a more straightforward and understandable way, you will not only make it more accessible to a wider audience but also increase the chances of your research being appreciated and utilized by others in your field.
Thank you for considering my suggestion, and I wish you the best of luck in your research endeavors.
Complicated sentence stucture, specialty in abstract paper.
Author Response
Reviewer 2
I hope this message finds you well.
Having carefully reviewed your work, it is evident that your findings hold significant value and have the potential to make a meaningful impact in your field. Furthermore, your research not only contributes to the existing body of knowledge but also offers practical solutions and recommendations that can be implemented to address specific challenges or improve current practices.
I recently had the opportunity to read the manuscript, and I wanted to provide you with some feedback regarding its clarity and comprehensibility. While reading the abstract, I found that the pattern of writing used made it quite challenging to grasp the overall theme and purpose of your paper. The language and structure employed seem to create a barrier to understanding, hindering readers from fully comprehending the essence of your work.
With the aim of improving the accessibility of your research, I would highly recommend revisiting the abstract and considering a simpler and more straightforward writing style. By presenting the main ideas and concepts in a clear and concise manner, you can ensure that readers can easily grasp the key points and understand the significance of your findings.
Remember that effective communication plays a crucial role in disseminating knowledge, and simplifying complex ideas can greatly enhance the impact of your work. By editing and describing the abstract in a more straightforward and understandable way, you will not only make it more accessible to a wider audience but also increase the chances of your research being appreciated and utilized by others in your field.
Thank you for considering my suggestion, and I wish you the best of luck in your research endeavors.
Author response:
We thank this reviewer for considering our paper and for their thoughtful comments on written language. We have taken these comments seriously and recognise that our writing style may need to be simplified. The senior author’s writing was based on standard guides such as Fowler’s Modern English Usage and Plain Words, the latter in recent editions, but we recognise that our style may have lapsed into forms more complex than needed to make the account readable, especially by those for whom English is not the first language. Therefore we have considered and amended the Abstract and simplified the written text at several points throughout the paper.
Reviewer 3 Report
Research is relevant and interesting.
1. No research hypothesis is presented in the introduction;
2. Methods of statistical analysis need to be described more clearly;
3. I recommend not to use old literature sources;
4. I recommend agrotechnologies of the experiment to present in the tabale;
5. I suggest providing photos of the experiment;
6. I recommend to provide more detailed conclusions.
Author Response
Reviewer 3
Research is relevant and interesting.
- No research hypothesis is presented in the introduction;
- Methods of statistical analysis need to be described more clearly;
- I recommend not to use old literature sources;
- I recommend agrotechnologies of the experiment to present in the tabale;
- I suggest providing photos of the experiment;
- I recommend to provide more detailed conclusions.
Author response:
The authors appreciate the constructive comments by the reviewer and have responded as follows:
- We intended to present the main aims and hypotheses as a set of questions at the end of the Introduction but have now rephrased them.
- The section on statistics has been expanded to describe how the main approaches were carried out.
- Old literature sources. We referred to some of the older literature, for example (Bennett 1935) since we felt it important to recognise that the problem of soil erosion following intensification of land use was known a century ago but has not been acted on in many parts of the world. We also refer to much published work in the 1990s but that also is highly relevant to the overall progression of knowledge and of the argument in the paper. This older literature is balanced by references to much new work published in the last few years. We would prefer to keep these references in the paper but have checked all reference to ensure that they are necessary to confirm the statements made. We have also updated the Statistical Methods by Bailey to a more recent edition (2012) and given a doi number.
- Thank you for this suggestion. We have constructed a Table to summarise the range of cropping systems as indicated and have located this in Supplementary Material Table S2.
- Similarly, some photographs of the crops are included in Supplementary Material.
- A Conclusions section is optional but we decided to include it. In response to your suggestion we have revised it to give more detail.
Reviewer 4 Report
The manuscript presents many interesting results and analyzes that are presented in a very intelligent way. However, in my opinion, it lacks important information on the studied species and forms of cereals.
An important part of the manuscript includes a comparison of the yield components and yield potential of barley and wheat. However, the authors did not specify which barley and wheat species were studied.
In my opinion, an uncorrect comparison of the yield elements of spring barley and winter wheat was made. This is inappropriate due to the different structure of the yielding elements of individual plants.
If one of the objectives of the research was the estimation of "technological maximum yield and the yield gap through grain number and mass", then the following can be compared:
Spring barley vs Spring wheat
Spring barley vs Winter barley
Winter barley vs Winter wheat
Spring wheat vs Winter wheat
But the manuscript presents a comparison of spring barley with winter wheat, which leads to wrong conclusions regarding the value of yield elements and the difference in the yield potential of the studied cereal species.
In addition, in my opinion, the research methodology, which in part 4.1 contains elements typical of Introduction, is incorrectly presented.
Other notes:
Line 516 - Authors wrote: “Samples of soil, crop and weeds were taken at three sample points...” whereas in lines 531-532 “no weed matter was found in any of the nine quadrats.”
Line 534-536: “To obtain the components of yield, six or more cereal ears were removed from each sample before drying and dissected to record the potential number of grain sites per ear, determined by the number of spikelets and florets, the actual number of grains per ear,..” – six ears is not enough to determine correctly the number of grains per ear on production field.
Conclusions refer to what was studied in the presented work to a very small extent.
Author Response
Reviewer 4
The manuscript presents many interesting results and analyzes that are presented in a very intelligent way. However, in my opinion, it lacks important information on the studied species and forms of cereals.
Authors response (AR): We thank the reviewer for the rigorous examination of the paper and appreciate your insights. We have indicated below our responses to your critique and recommendations.
An important part of the manuscript includes a comparison of the yield components and yield potential of barley and wheat. However, the authors did not specify which barley and wheat species were studied.
AR: Our omission: species and intended markets of the wheat and barley sampled are now given in Methods. Species names are given in the Introduction also.
In my opinion, an uncorrect comparison of the yield elements of spring barley and winter wheat was made. This is inappropriate due to the different structure of the yielding elements of individual plants.
AR: we discuss this important point below.
If one of the objectives of the research was the estimation of "technological maximum yield and the yield gap through grain number and mass", then the following can be compared:
Spring barley vs Spring wheat
Spring barley vs Winter barley
Winter barley vs Winter wheat
Spring wheat vs Winter wheat
But the manuscript presents a comparison of spring barley with winter wheat, which leads to wrong conclusions regarding the value of yield elements and the difference in the yield potential of the studied cereal species.
AR: We thank the reviewer for raising these issues on barley and wheat, their yield components and the estimates of technological maximum yield. We respond to these points in a separate section below.
In addition, in my opinion, the research methodology, which in part 4.1 contains elements typical of Introduction, is incorrectly presented.
AR: we have considered section 4.1 and moved some of it to the Introduction.
Other notes:
Line 516 - Authors wrote: “Samples of soil, crop and weeds were taken at three sample points...” whereas in lines 531-532 “no weed matter was found in any of the nine quadrats.”
AR: Thanks for pointing out this potential confusion. We have changed the text to indicate that the same sampling procedure was carried out at all loci but that weeds were not found at any of the loci at a proportion of the sites.
Line 534-536: “To obtain the components of yield, six or more cereal ears were removed from each sample before drying and dissected to record the potential number of grain sites per ear, determined by the number of spikelets and florets, the actual number of grains per ear,..” – six ears is not enough to determine correctly the number of grains per ear on production field.
AR: Apologies for confusion here: six ears were dissected per sample locus, not per field. We found that six per locus multiplied by number of sample loci per field was sufficient for an accurate estimate of yield components. Also, the main aim of the analysis was not primarily to compare yield between fields but to examine the overall relations among population density, yield, and yield components as in Figs 5 and 6. The text has been modified to explain.
Conclusions refer to what was studied in the presented work to a very small extent.
AR: we have revised the Conclusions.
Authors responses to comments on wheat, barley, yield components and technological maximum yield.
We appreciate the reviewer’s insightful comments on these points and have revised the paper accordingly. Our arguments are as follows:
- Comparison of barley and wheat. We attempt to counter the reviewer’s position here. If the species differed fundamentally in head structure – as for example between maize, pearl millet and barley – yes, there would be difficulty in comparing species in terms of the same reproductive attributes. However, we feel it legitimate to compare C3 species, wheat and barley, using the same head/ear components because the species growing at the sites in the study had a very similar basic structure. The head of each species consists of a rachis (or spike) that holds reproductive sites in the form of spikelets, which themselves contain florets in which grains may develop. When the spikelets are removed, the rachis in barley and wheat is similar in appearance. The rachis changes its angle at each of the ‘notches’ that support spikelets. The two species had a similar number of notches, or spikelet sites (about 20 per head, but slightly more in barley that wheat). In two-row barley, two of the reproductive units at any notch are barren and the remaining one contains only one grain. In wheat, all three units can develop and each can hold more than one grain. The individual grain weights cited in the paper averaged over all samples and sites were almost the same. The main difference between the species was in the number and arrangement of fertile florets (= grains) of which there are about twice a many in winter wheat. This basic similarity leads to the unifying relation in Fig. 6 between grain number and yield.
- Technological maximum. In the Introduction, the yields in separate studies of wheat and barley (Fig. 1) are compared by a common analysis in terms of basal, current mean, technological and pedo-climatic yields. This concept of a technological maximum has been found useful in many situations and is commonly estimated as the highest yield that can be obtained in highly controlled field experiments or even as the highest yield among a survey of farms and fields. It is a guide therefore as to what a field might attain in a given climate if the agronomy was the best it could be. However, we agree that may be some confusion in the Results due to our emphasis on the potential in wheat. We will remove that emphasis and deal with both species similarly. However, we will also argue in the Discussion that to raise yields (and nitrogen inputs) substantially would include moving to mainly winter crops that have the potential for a high grain number.
- Further comparison of winter/spring, barley/wheat. While comparison of the various forms as indicated by the reviewer would be possible in a controlled field experiment, the aim of the work reported here was to assess the position as it is in commercial agriculture, 50 years after intensification began. We sampled the two cereal crops that cover the largest area. These would have been different 50 years ago (oats and barley) and may be different in the future. We accept that comparisons as indicated by the reviewer would contribute to defining yield potentials more widely, but we maintain that it is legitimate to present data on wheat and barley for our stated purposes.
- Actions: 1) Revise the text in Results/Discussion to place less emphasis on ‘technological maximum’ as usually defined and concentrate more on the series of higher yields that could be achievable by manipulating grain components, together with estimating the extra nitrogen needed to support those yields. 2) Emphasise that winter wheat and spring barley are measured and compared because they are the current crops most widely grown, and that the comparison is valid because their underlying structure is similar. 3) Give equal weight to estimating raised yield and nitrogen for both spring barley and winter wheat.
Once again we thank the reviewer for your detailed critique which has helped us improve our presentation and arguments.
Round 2
Reviewer 4 Report
The manuscript has been substantially improved and most of my suggestions have been taken into account.
However, the conclusions, which in my opinion should refer to the results of the research carried out by the authors, have not been corrected. So, in their current form, they go far beyond these results.
In addition, I suggest in the discussion to justify the increase in yield over the period of 50 years not only by biological and agrotechnical progress, but also (partially) by changing spring cereals (less productive) to winter cereals (more productive).
Author Response
Reviewer 4, second review:
The manuscript has been substantially improved and most of my suggestions have been taken into account.
Authors’ response (AR): we again appreciate the time and expertise offered by reviewer 4 and respond as follows to their comments on Revision 1.
However, the conclusions, which in my opinion should refer to the results of the research carried out by the authors, have not been corrected. So, in their current form, they go far beyond these results.
AR: The Conclusions section was re-written for Revision 1 in the light of Reviewer 4’s first set of comments, but we note that Reviewer 4 thinks they still go beyond remit. We believe that Conclusions should be able to cover points made in the Discussion as well as in the Results. In Revision 2, the text has been modified in response to Reviewer 4’s comments, but the main points of the Discussion are little changed from those in Revision 1. Paragraph 1 draws on material in the Results sections 2.1, 2.2 and 2.3 and Figures 2, 3 and 4 on C and N in soil and crops, and to a lesser degree to material in Discussion section 3.1. The first part of paragraph 2 relates to information in Results, mainly in sections 2.4 and 2.5, and also in the Discussion at section 3.2. The second part of paragraph 2 then expands the coverage to include material introduced in the Discussion, but which points to related, published work in the same region as the present study. Combining the current study with related work enables us to widen the significance of the present study.
In addition, I suggest in the discussion to justify the increase in yield over the period of 50 years not only by biological and agrotechnical progress, but also (partially) by changing spring cereals (less productive) to winter cereals (more productive).
AR: to clarify, sentences have been added to (a) the Discussion in the second paragraph of Section 3.2 to confirm this point and (b) to the results at section 2.5 to further compare (and distinguish) barley and wheat.